# Time to death from cervical cancer and predictors among cervical cancer patients in Felege Hiwot Comprehensive Specialized Hospital, North West Ethiopia: Facility-based retrospective follow-up study

**Andamlak Eskale Mebratie[1], Nurilign Abebe Moges[2], Belsity Temesgen Meselu[3], Misganaw Fikrie Melesse** [3] *

1 Department of Epidemiology, Sidama Region Health Bureau, Awassa, Ethiopia, 2 Department of Public Health, Debre Markos University, Debre Markos, Ethiopia, 3 Department of Midwifery, Debre Markos University, Debre Markos, Ethiopia

* misganawfikrie@gmail.com

**Data Availability Statement:** All relevant data are within the paper.

## Abstract

### Introduction

A sexually transmitted virus called the Human Papillomavirus is responsible for more than 99% of cervical cancer cases and its precursors. In 2019, the median survival time of cervical cancer patients at 5 years was 37 months. The survival time and predictors of death from cervical cancer vary in different study settings. This study was aimed to assess the time to death and to identify the major predictors of death of cervical cancer patients in Felege Hiwot Comprehensive Specialized Hospital.

### Methods

A facility-based retrospective follow-up study was conducted among 422 randomly selected cervical cancer patients in Felege Hiwot Comprehensive Specialized Hospital from 25th June 2017 to 31st March 2021. Data were extracted from the sampled patient charts by using a structured checklist which was prepared in an English version. Data were coded and then entered, edited, and cleaned using EPI-data 3.1 and exported to STATA14.2 statistical software for analysis. Frequencies and proportions were used to describe the study population with relevant variables and were presented using tables, pie charts, and graphs. Kaplan Meier and life table were used to describe the restricted mean survival time and the overall survival rates. Differences in survival among different variables were compared using the log-rank test. The assumption of proportional hazard was checked using Schoenfeld residual test. Variables having a P-value > 0.05 were considered as fulfilling the assumption. Variables with a significance level below 0.2 in the bivariable Cox regression model were included in a multivariable Cox regression model analysis, where Variables with a p-value < 0.05 were considered to be statistically significant at a 95% confidence interval. Model fitness was checked by Cox-Snell residual.

**Funding:** The author(s) received no specific funding for this work.

**Competing interests:** The authors have declared that no competing interests exist.

**Abbreviations:** AHR, Adjusted Hazard Ratio; CCI, Carlson comorbidity Index; CIN, Cervical Intraepithelial Neoplasia; DMU, Debre Markos University; FIGO, International Federation of Gynecologists and Obstetricians, FHCSH: Felege Hiwot Comprehensive Specialized Hospital; HPV, Human Papillomavirus; ICCs, Invasive Cervical Cancers; MOH, Ministry of Health; RMST, Restricted Mean Survival Time; SSA, Sub Saharan Africa; TASH, Tikur Anbessa Specialized Hospital.

## Results

The mean follow up time of this cohort was 27.66 (CI: 26.96, 28.36) months, and the restricted mean survival time of cervical cancer patients in this study was 40.21 (95% CI: 38.95, 41.47) months. Being FIGO stage IV [AHR = 6.10, 95% CI: 2.18, 16.90)], having adenocarcinoma [AHR = 3.12, 95% CI: 1.34, 7.28)], having co-morbidity [AHR = 2.57, 95% CI: 1.29, 5.11)], and being initiated with radiotherapy [AHR = 4.57, 95% CI: 1.60, 13.06)] were a significant predictors of death from cervical cancer.

## Conclusion

The restricted mean survival time of cervical cancer patients in this study was 40.21 months. Marital status, type of tumor histology, stage of disease, type of treatment initiated, and presence of co-morbidity were significant predictors of death for cervical cancer. Treatment of comorbidities in the early stage of cervical cancer plays a key role in maximizing the survival time of cervical cancer patients.

## Introduction

Cervical cancer (CC) is a malignant neoplasm from cells originating in the cervix [1] which is an essential cause of mortality amongst women [2]. Cervical cancer accounts for 7.5% of all female cancer deaths globally peaking between the ages of 35–65 years and not only takes the life of young women but devastates families with young children along the way [2]. The magnitude and distribution of cervical cancer vary across countries in the world [3]. In 2020, the estimated cervical cancer mortality rate across all 78 low-income and lower-middle-income countries were 13·2 per 100 000 women [4].

Cervical cancer has several consequences including mortality varied 18-folds between the different regions of the world [5] being a major public health problem around the globe [3], particularly in less-resourced countries [6]. Cancer and its treatment can cause substantial disruptions in school and career, as well as changes in functioning and appearance, leading to further challenges in resuming daily-life activities [5]. Cervical cancer has become a severe force to be reckoned with a huge challenge for the health care systems of Africa [7].

The burden of cancer is increasing in Africa because of the aging and growth of the population as well as the increased prevalence of risk factors associated with economic transition, including smoking, obesity, physical inactivity, and reproductive behaviors [8]. Despite this growing cancer burden, cancer continues to receive a relatively low public health priority in Africa, largely because of limited resources and other pressing public health problems [8].

Patients affected by cervical cancer survive longer than 5 years in less than 50% of women in less-developed countries [3]. The survival of cervical cancer patients in economically advanced regions is about 66% at 5 years [4]. Large variation in survival is observed among developing countries due to variations in stages of presentation and accessibility to diagnostic and treatment services [9]. A 4-year overview of treated patients with CC in Ethiopia concluded that due to limited treatment availability, the 1 and 2-year overall survival probabilities were 90.4% and 73.6% with a relatively high proportion of patients who survived 2 years, which is less than the survival time stated by the study conducted at TARH [9]. This was also considerably high (74%) compared with data from African cancer registries [10]. Socioeconomic, demographic, and medical variables like age, marital status, origin, occupation, histological type are known to be associated with survival [11]

Many cultural factors contribute to the acquisition of HPV and its progression to cancer which include high parity, early marriage, multiple sexual partners, and diseases that reduce immunity status [12]. These were also among the possible predictors associated with higher cervical cancer mortality among women in addition to early age at first sexual intercourse, advanced cervical cancer disease condition, smoking, and use of oral contraceptives [13] [14]. It was observed that retired and divorced women had lower survival rates than other women [11].

So far, only a few kinds of the literature revealed the lower overall survival probabilities of cervical cancer patients in Ethiopia when compared with those of high- and middle-income countries [9]. There was a gap in information about the survival of cervical cancer patients in the study area due to the lack of recorded materials in the study area to compare the survival of CC patients with the research findings at the national level. Because the prognostic predictors of CC death vary in different studies [12]. The aim of this study was intended to identify the current mortality predictors of CC. By doing so, this study could help in filling information gaps related to the death of women from cervical cancer in the study area.

## Methods

### Study area and period

The study was conducted at Felege Hiwot Comprehensive Specialized Hospital (FHCSH), in Bahir Dar Town, North West Ethiopia. Bahir Dar Town, the capital city of the Amhara region, is found 565 Kilometers far from Addis Ababa, the capital city of Ethiopia, and 265 kilometers far from Debre Markos, East Gojjam Zone. FHCSH serves a total of 12 million populations. The hospital has 18 beds for inpatient admissions of cancer patients. The oncology center of FHCSH was established in June 2017. It is providing cervical cancer screening, early detection, prevention, and treatment including chemotherapy, surgical treatment, and palliative care services for cervical cancer patients. The study was conducted from 25th 7 March 2017 to 31st March 2021.

### Study design

A facility-based retrospective follow-up study was conducted at the oncology center of FHCSH.

### Study participants

All women with cervical cancer attended the oncology center of FHCSH from 25th June 2017 to 31st March 2021.

### Sample size calculation

The required sample size was computed using the Freedman method of proportional event allocation, as follows.

Sample size (n) = $\frac{number\ of\ events}{Probability\ of\ event}$ [15]

Number of events = $\frac{(Z\alpha/2 + Z\beta)^2}{pq(logHR)^2}$ [15]

Probability of the event = 1-(ps1 (t) +qs2 (t))

where n is the sample size; $z_{a/2}$ is a significant level of α of 5%, which is 1.96; β is the power (80%), p is the proportion of population allocated for the first group, q proportion of population allocated for the second group, $s1_{(t)}$ is survival function at time t1, $s_2$ is survival function at time t2 and HR is the hazard ratio of each variable which are taken from similar previous

**Table 1. Sample size to identify predictors of death among women with cervical cancer admitted at FHCSH, North West Ethiopia, 2021.**

| Variable | AHR | 95% CI | proportion of withdrawing | P(event) | event | required sample size | Reference |
|---|---|---|---|---|---|---|---|
| Advanced Age | 5.99 | (2.1, 17.08) | 0.1 | 0.69 | 22 | 32 | [9] |
| substance use | 1.56 | (1.09, 2.22) | 0.1 | 0.52 | 220 | 422 | [9] |
| Comorbidity | 2.60 | (1.67, 4.04) | 0.1 | 0.6 | 54 | 90 | [16] |
| baseline anemia | 3.7 | (1.8, 7.5) | 0.1 | 0.64 | 32 | 50 | [17] |
| Tumor type | 2.85 | (0.99, 8.21) | 0.1 | 0.61 | 46 | 76 | [13] |
| advanced FIGO stage | 3.3 | (1.2, 8.9) | 0.1 | 0.63 | 38 | 60 | [17] |
| Treatment modality | 0.45 | (0.19, 1.12) | 0.1 | 0.34 | 74 | 212 | [17] |

studies. In the freedman principle approach, the equal allocation between these groups was assumed. That means p = q = 0.5. Stata version 14.2 was used to calculate the required sample size by using the "power" command (power log-rank 0.5, hratio(1.56) power (0.9) wdprob (0.1)) of STATA. The hazard ratios of seven variables that were statistically significant from the previous articles were used in the calculation of the sample size for this study. The largest sample size obtained by the above command was 422. That was taken as the final required sample size for the study (**Table 1**)

## Sampling technique

Simple random sampling was used to select study participants. First of all, the total number of women with cervical cancer who were on follow-up from 25th June 2017 to 31st March 2021 in FHCSH was counted and recorded from the patient registration book. The total count of cervical cancer patients in the specified years of enrolment period was 764. The total unit of the frame was thus from 1 to 764. These recorded data were entered into computer excel and made randomized. Then the randomized medical record numbers were sorted so that the desired number of subjects (n = 422) were taken top to down from the list.

## Variables

**Dependent variable.** Time to death.

**Independent variables.** *Socio-demographic predictors*. **Age at diagnosis, residence, marital status, Substance use, parity/ number of children,**

*Pathological and clinical predictors*. Stage of cervical cancer at presentation, histological type of the tumor, baseline anemia, types of baseline co-morbidity.

*Treatment-related predictors*. Type of treatment/ Chemotherapy, Radiation, surgery, a combination of treatments, treatment cycle, and treatment duration/ in months.

**Operational definition.** *Survival time*. The time calculated from the initiation of the treatment of cervical cancer to dead or censored.

*Event*. Death of cervical cancer patients during the entry date and closing date to follow-up.

*Censored*. Patients, who lost to follow-ups, do not die up to the study period, and those transferred to different care units during the study.

*Comorbidity*. The presence of any conditions (mentioned in the Carlson comorbidity Index [18] other than cervical cancer at diagnosis which was designated as "yes" in the checklist.

*Substance use*. Patients who ever used one or more substances (cigarette, caffeine, chat, and alcohol).

*Entry date to follow-up*. The entry date was the first date of a clear diagnosis of cervical cancer (25th June 2017).

*Closing date to follow-up*. The closing date was the ending date to follow up (31st March 2021).

*Survival status*. Survival status was defined as the outcome of patients, which was classified as censored or death from the patient's clinical data file.

*Time to death*. Time to death was calculated at the time between the dates of admission with a clear diagnosis of cervical cancer to the date of death (in months).

*The stage at diagnosis*. The revised FIGO staging for carcinoma of the cervix was used [19].

## Data collection procedures, and quality control

Secondary data was collected using a structured standard data abstraction checklist. The checklist was prepared in English from previous studies [9, 12, 17, 20, 21]. The checklist contained five parts to include necessary evidence from patients' charts: (1). socio-demographic predictors (2). Treatment-related predictors, (3). Pathologic and clinical predictors including the follow-up history.

The data was extracted by three nurses who work in the oncology unit and they were supervised by one trained public health specialist. The training was given to the supervisor and data extractors on sampling procedures, techniques of the data extraction process for one day, and any doubt in the structured checklist was clarified.

## Data processing and analysis

Data was entered using EPI-data 3.1 and exported to STATA14.2 statistical software for coding, entering, editing, cleaning, and analysis. Frequencies, proportions, and descriptive statistics were used to describe the study population with relevant variables and were presented using tables and graphs. Kaplan Meier and life table were used to describe the overall survival rates and restricted mean survival time. Since the median survival time for the cohorts of cervical cancer patients in this study was undetermined, the restricted mean survival time was calculated. Differences in survival among different variables were compared using the log-rank test. Before running the Cox regression model, the assumption of proportional hazard was checked using Schoenfeld residual test, and variables having a P-value $> 0.05$ were considered as fulfilling the assumption. Variables with a significance level below 0.2 in the bivariable Cox regression model were included in a multivariable Cox regression model analysis. Variables in the multivariable Cox model with a p-value $< 0.05$ were considered to have actual interference with the survival of the patients with a 95% confidence interval. Model fitness was checked by Cox-Snell residual.

## Ethical consideration

Ethical approval for this study was obtained from the Ethical Review Committee of DMU, College of Health Sciences, and Department of public health. The department of public health wrote a support letter for FHCSH. After getting permission from FHCSH, the required data was obtained from the patient charts.

## Results

### Socio-Demographic characteristics of the study participants

From 764 cervical cancer patients in the oncology center of FHCSH, about 422 were eligible samples for this study during the study period. The mean age of study participants was 50.11 years with SD ± 12.7 ranging from 25 to 78 years. Two-hundred ninety-three (69.43%) of

**Table 2. Socio-demographic characteristics of cervical cancer patients in Felege Hiwot Comprehensive Specialized Hospital Oncology Center, Ethiopia (N = 422, March 2021).**

| Covariates | Category of covariates/ response | Status at last contact | | Total Frequency (%) |
|---|---|---|---|---|
| | | Death No (%) | Censored No (%) | |
| Age at diagnosis | <30 years | 5 (17.86) | 23 (82.14) | 28 (6.64) |
| | 30–39 years | 6 (11.32) | 47 (88.68) | 53 (12.56) |
| | 40–49 years | 7 (5.56) | 119 (94.44) | 126 (29.86) |
| | 50–59 years | 10 (9.18) | 99 (90.82) | 109 (25.83) |
| | > = 60 years | 15 (14.15) | 91(85.85) | 106 (25.12) |
| Residence | Urban | 7 (5.43) | 122 (94.57) | 129 (30.57) |
| | Rural | 36 (12.29) | 257 (87.71) | 293 (69.43) |
| Marital status | Married | 16 (5.84) | 258 (94.16) | 274 (64.93) |
| | Unmarried | 10 (27.03) | 27 (72.97) | 37 (8.77) |
| | Divorced | 9 (15.52) | 49 (84.45) | 58 (13.74) |
| | Widowed | 8 (15.09) | 45 (84.91) | 53 (12.56) |
| Substance use | Yes | 5 (15.15) | 28 (84.85) | 33 (7.82) |
| | No | 38 (9.78) | 351 (90.23) | 389 (92.18) |
| Number of children | No child | 12 (37.5) | 20 (62.5) | 32 (7.58) |
| | One | 9 (7.32) | 114 (92.68) | 123 (29.15) |
| | Two | 9 (6.47) | 130 (93.53) | 139 (32.94) |
| | Three or more than three | 13 (10.16) | 115 (89.84) | 128 (30.33) |

participants were rural dwellers. Nearly two-thirds (64.93%) of the study participants were married. (**Table 2**).

## Survival status of cervical cancer patients, at last, follow up

Of the total 422 eligible samples of cervical cancer patients in the oncology center of FHCSH who were followed for 46 months, about 379 (89.8%) were censored and 43 (10.2%) developed the event of interest, i.e died. More than half (56.2%) of the censoring was alive up to the end of the study period. Those who were transferred out to other facilities were 90 (23.75%) and lost to follow-up were 76 (20.05%) (**Fig 1**).

## Clinical and histopathological characteristics

Among 422 cervical cancer patients, nearly one-third (33.89%) of them were presented at advanced stages (IV), three hundred seventy-one (87.91%) had squamous cell carcinoma, one hundred eighty-two (43.13%) were anemic during the presentation. Nearly one-third (30.80%) had comorbidity and of those who had comorbidity, nearly half of them (48%) were HIV positives (**Table 3**).

## Treatment-Related characteristics

Of a total of 422 study subjects, nearly two-thirds (63.98%) were initiated with a combination of chemotherapy, radiation, or/ and surgical treatments. Among these, one hundred eighty-five (43.84%) took chemo-radiation treatment only. About one hundred forty-eight (35.07%) took treatments without combination. Among two hundred sixty-five (62.80%) patients who took radiotherapy, one hundred forty-eight (35.07%) patients took radical radiotherapy. One hundred eighty-four (43.60%) of patients had four or more than four chemotherapy cycles (**Table 4**)

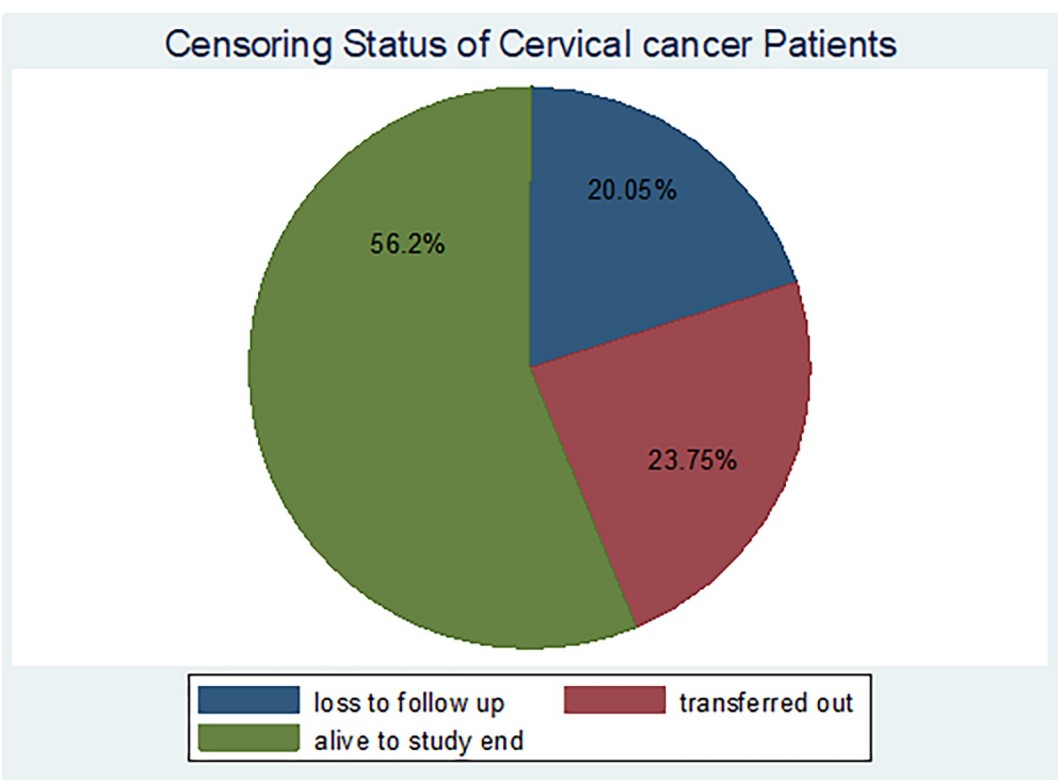

**Fig 1. Censoring status of cervical cancer patients in Felege Hiwot Comprehensive Specialized Hospital oncology center, Ethiopia (n = 422, March 2021).**

**Table 3. Clinical and histopathological characteristics of cervical cancer patients in Felege Hiwot Comprehensive Specialized Hospital Oncology Center, Ethiopia (N = 422, March 2021).**

| Covariates | Category of covariates/ response | Status at last contact | | Total No (%) |
|---|---|---|---|---|
| | | Death No (%) | Censored No (%) | |
| FIGO stage | Stage I (IA-IIA) | 8 (11.43) | 62 (88.57) | 70 (16.59) |
| | Stage II (IIB-IIIA) | 7 (8.33) | 77 (91.67) | 84 (19.91) |
| | Stage III (IIIB-IVA) | 10 (8) | 115 (92) | 125 (29.62) |
| | Stage IV (IVB) | 18 (12.59) | 125 (87.41) | 143 (33.89) |
| Histopathology | Squamous cell | 30 (8.09) | 341 (91.91) | 371 (87.91) |
| | Adenocarcinoma | 13 (25.5) | 38 (74.5) | 51 (12.09) |
| Baseline Anemic status | Yes | 23 (12.64) | 159 (87.36) | 182 (43.13) |
| | No | 20 (8.33) | 220 (91.67) | 240 (56.87) |
| Baseline Comorbidity | Yes | 19 (14.62) | 111 (85.38) | 130 (30.81) |
| | No | 24 (8.22) | 268 (91.78) | 292 (69.19) |
| Types of comorbidity | HIV | 9 (11.25) | 71 (88.75) | 80 (18.96) |
| | Hypertension | 6 (13.33) | 39 (86.67) | 45 (10.67) |
| | Diabetic Mellitus | 2 (10.53) | 17 (89.47) | 19 (4.5) |
| | Other* | 2 (8.7) | 21 (91.3) | 23 (5.45) |

* The disease conditions listed in the CCI except for CC, HIV, hypertension, and diabetic Mellitus [22]

**Table 4. Treatment-Related characteristics of cervical cancer patients in Felege Hiwot Comprehensive Specialized Hospital Oncology Center, Ethiopia (n = 422, March 2021).**

| Covariates | Category of covariates/ response | Status, at last, follow up | | Total (%) |
|---|---|---|---|---|
| | | Death No (%) | Censored No (%) | |
| Treatment initiated | Chemotherapy | 8 (6.78) | 110 (93.22) | 118 (27.96) |
| | Radiation | 8 (34.78) | 15 (65.22) | 23 (5.45) |
| | Surgery | 6 (54.55) | 5 (45.45) | 11 (2.61) |
| | Combination of treatments | 21 (7.78) | 249 (92.22) | 270 (63.98) |
| Combination of treatments | Surgery and chemo | 3 (9.68) | 28 (90.32) | 31 (7.35) |
| | Chemoradiation only | 16 (8.65) | 169 (91.35) | 185 (43.84) |
| | surgery & radiotherapy | 3 (6.12) | 46 (93.88) | 49 (11.61) |
| | Surgery + chemotherapy + radiotherapy | 2 (22.22) | 7 (77.78) | 9 (2.13) |
| | No combined treatments | 19 (12.84) | 129 (87.16) | 148 (35.07) |
| Chemotherapy cycles | No chemotherapy | 14 (64.04) | 64 (35.96) | 78 (18.48) |
| | First cycle | 5 (17.86) | 23 (82.14) | 28 (6.64) |
| | Second cycle | 6 (10) | 54 (90) | 60 (14.22) |
| | Third cycle | 8 (11.11) | 64 (88.89) | 72 (17.06) |
| | Fourth or more cycles | 10 (5.43) | 174 (94.56) | 184 (43.60) |
| Aim of radiation treatment | Radical radiotherapy | 16 (10.81) | 132 (89.19) | 148 (35.07) |
| | Palliative care | 12 (10.26) | 105 (89.74) | 117 (27.73) |
| | No Radiotherapy | 15 (9.55) | 142 (90.45) | 157 (37.20) |

## Survival among group of cervical cancer patients

Statistical difference in survival time between different categories of covariates was tested using the Log-rank test. It was found that there is a significant difference in survival experience among the categories of residential address (Fig 2), marital status, the number of children, stage of cervical cancer, type of histology of the tumor, type of treatment initiated, presence of co-morbidity, and the number of chemotherapy cycles at p-value < 0.05.

The restricted mean survival time for those who married was longer (41.86 months) (95% CI: (40.42, 43.32)) than those who were unmarried, divorced, and widowed. The restricted mean survival time for those who had FIGO stage II or III at baseline was longer than those in stage IV (36.57 months) (95% CI:(34.30, 38.83)) compared to those who had stage **I (Fig 3).**

The restricted mean survival time for those who had squamous cell carcinoma was longer (41.05 months; 95% CI: (39.75, 42.36)) than for those who had adenocarcinoma (34.90) (95% CI: (32.11, 37.70)). The restricted mean survival time for cervical cancer patients with co-morbid illness was shorter (38.14 months; 95% CI: (36.05, 40.22)) than for those who had no co-morbidity. The restricted mean survival times for cervical cancer patients who were treated with chemotherapy, radiation therapy, and surgery, were longer than those who had been treated with the combination of the treatments (40.83 months; 95% CI: (39.28, 42.38)) (Fig 4)

Cervical cancer cases that had husbands had a higher cumulative survival rate (64.10%) than those who were not married (18.24%), divorced (48.88%), and widowed (40.27%). (**p = 0.00**). For cervical cancer cases diagnosed at an early stage (I&II), the cumulative survival rate was 61.89% and 72.36% respectively, and those cases diagnosed at an advanced stage (III&IV) were 4378 and 9.18% respectively (**p = 0.01**). Cervical cancer patients that had squamous cell carcinoma and adenocarcinoma had cumulative survival rates of 58.71% and 12.38% respectively (**p = 0.00**).

Patients who had co-morbidity had a shorter cumulative survival rate (33.95%) than those who did not experience additional illness to cervical cancer (62.87%) (Fig 5).

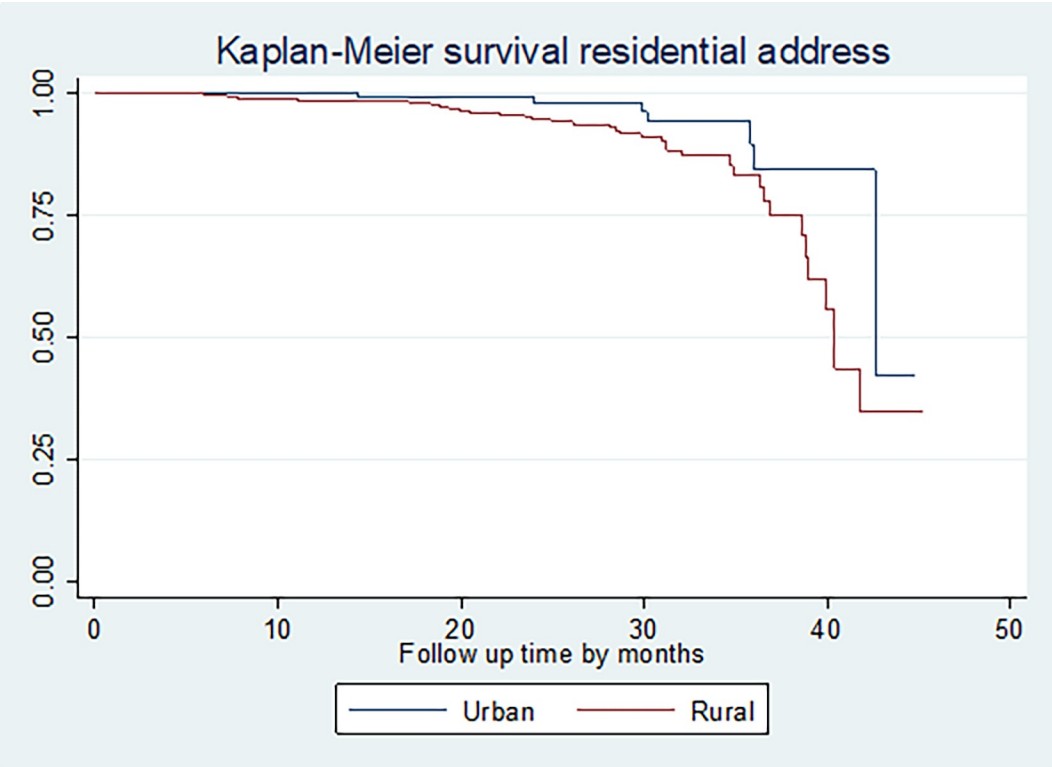

**Fig 2. The Kaplan-Meier survival curves comparing survival time of cervical cancer patients by residential address in Felege Hiwot Comprehensive Specialized Hospital Oncology Center, Ethiopia (n = 422, by residence, n = 422, March 2021).**

Patients treated with chemotherapy had a higher cumulative survival rate (64.73%) than those treated with radiotherapy, surgery, or a combination of chemotherapy, radiotherapy, and/ surgery (25.17%, 16.40%, and 59.64%, respectively) (**Table 5**).

## Overall survival of cervical cancer patients

In this study, 422 cervical cancer patients were followed for 46 months. The mean follows up time and the restricted mean survival time of this cohort was 27.66 (CI: 26.96, 28.36) and 40.21 (95% CI: 38.95, 41.47) months, respectively. The overall estimated survival rate of cervical cancer patients was 53.15% at 46 months of follow-up. The estimated cumulative survival was 99.04%, 96.00%, 87.63% and 53.15% at 12, 24, 36 and 46 months respectively. The overall incidence of cervical cancer death among cervical cancer patients in this study was 44% per 1000 person-years for 972.8 years of observation. While the median survival time was undetermined because the largest observed analysis time was censored, the survivor function does not go to zero; in this case, the restricted mean is the best estimate of survival time [23] (**Fig 6**).

## Survival time predictors among cervical cancer patients

The independent variables were analyzed individually with the outcome variable and categories of age, several children, residential address, marital status, substance ever used, stages of cervical cancer disease, tumor type, condition of co-morbidity, and the types of treatments initiated were fitted at p-value < 0.2. Before running the Cox regression model, the assumption of proportional hazard was checked using Schoenfeld residual test (global test found to be

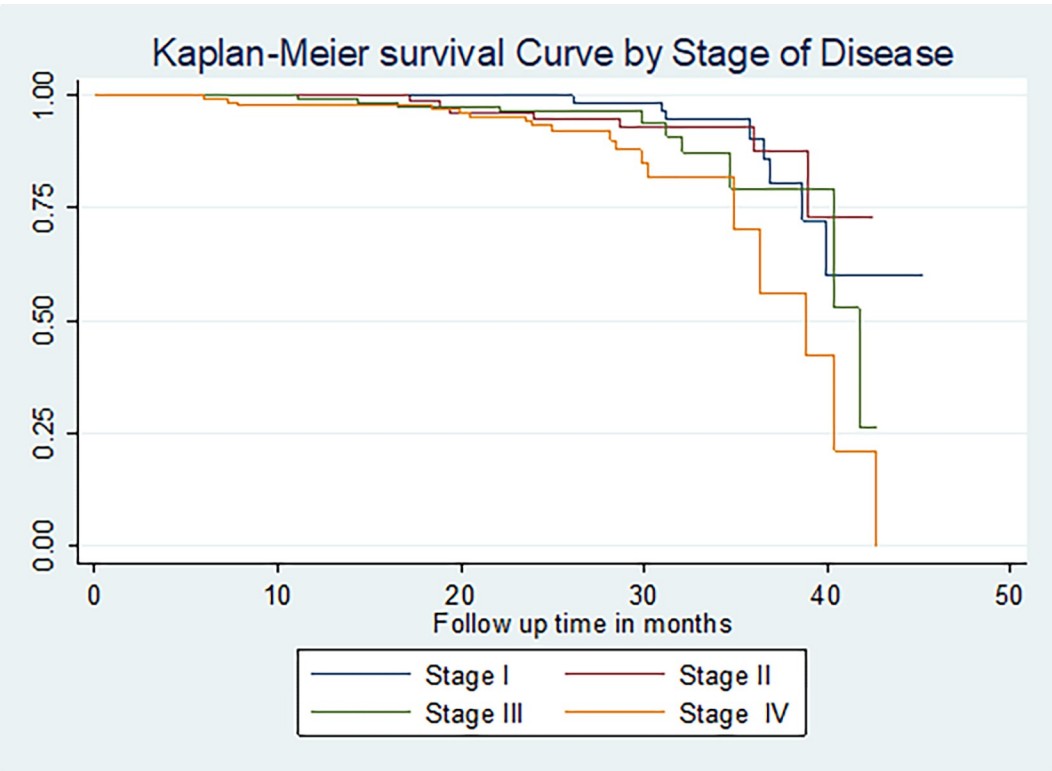

**Fig 3. The Kaplan-Meier survival curves comparing survival time of cervical cancer patients by stage of disease in Felege Hiwot Comprehensive Specialized Hospital Oncology Center, Ethiopia (N = 422, By Stage of Disease, N = 422, March 2021).**

0.61), the graphical method, and/ or the time-varying Covariates (TVC), and variables having P-value > 0.05 were considered as fulfilling the assumption (**Table 6**) (**Fig 7**).

The above **Fig 7** implies that the proportional-hazards assumption for types of histology group was not violated.

As shown in **Fig 8**, the proportional-hazards assumption for types of comorbidity group was not violated. In addition, by the test of proportional-hazards assumption by time-dependent variables, there was no significant evidence that suggested the assumption of proportional hazards was violated.

After the PH assumption was checked for no violations, the categories of age, number of children, residential address, marital status, substance ever used, stages of cervical cancer disease, tumor type, condition of co-morbidity, and the types of treatments initiated were included in the multivariate Cox regression model. The result of the multivariable analysis revealed that married women were 0.15 times [AHR = 0.15, 95% CI: 0.06, 0.37)], at less hazard of dying compared to unmarried ones. Women with advanced disease stage (stage IV) were 6.10 times [AHR = 6.10, 95% CI: 2.18, 16.90)] at high risk to die as compared to those with early stage of disease (FIGO stage I). Those women with adenocarcinoma were 3.12 times [AHR = 3.12, 95% CI: 1.34, 7.28)] at high risk to die than those with Squamous cell carcinoma. Those cervical cancer patients who had a co-morbid illness were 2.57 times [AHR = 2.57, 95% CI: 1.29, 5.11)] at high risk to die than those without co-morbidity. In addition, cervical cancer patients who had been treated with radiotherapy were 4.57 times [AHR = 4.57, 95% CI: 1.60, 13.06)] at a higher hazard to develop the event than those who have initiated a different combination of treatments (**Table 7**).

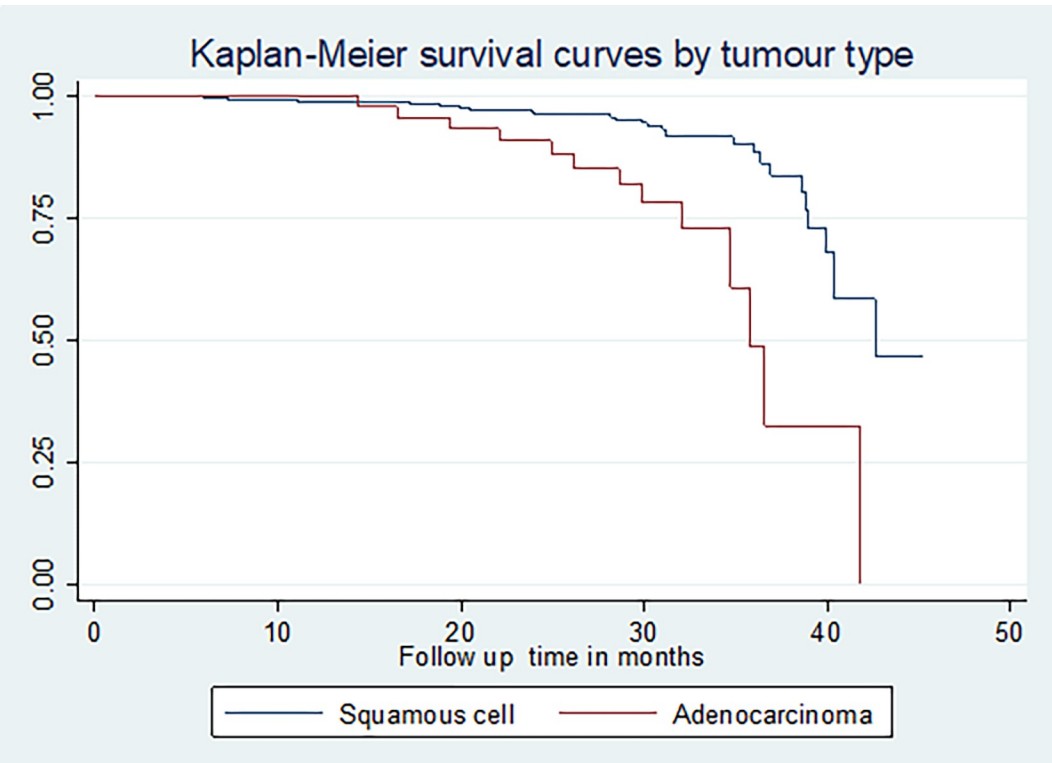

**Fig 4. The Kaplan-Meier survival curves comparing survival time of cervical cancer patients by tumor type in Felege Hiwot Comprehensive Specialized Hospital Oncology Center, Ethiopia (N = 422, By Type of Histology, N = 422, March 2021).**

Proportional Hazard Model fitness was checked by the Cox-Snell residual as shown in the following figure (**Fig 9**).

As shown in **Fig 9**, the function follows the 45-degree line very closely except for some large values of time, so that the final model fits the data well. It is very common for models with censored data to have some wiggling at large values of time and it is not something that should cause much concern.

## Discussion

This study was undertaken to assess the time to death of cervical cancer patients and the predictors of death among cervical cancer patients in FHCSH.

The median survival time of cervical cancer patients was undetermined; instead, the restricted mean survival time of cervical cancer patients was estimated and found to be 40.21 months. The overall incidence of cervical cancer death from 422 cervical cancer patients in this study was 44% per 1000 woman-years. The overall estimated survival rate of cervical cancer patients in this study was 53.15% at 46 months of follow-up.

The estimated cumulative survival in this study was 99.04%, 96.0%, 87.63%, and 53.15% at 12, 24, 36, and 46 months respectively. These were higher than the overall survival rates in Ethiopia (90.4% and 73.6% at 12 and 24-months) and in SSA (67.5% and 46.2% at 12 and 36 months, respectively) [24, 25]. This may be due to different study periods and settings, or individual patient conditions like patients' treatment adherence.

The cumulative survival of cervical cancer patients at stage IV in this study (9.18%) was lower than in previous studies in Ethiopia (20.03%), Nigeria (15%), and SSA (20.05%) but

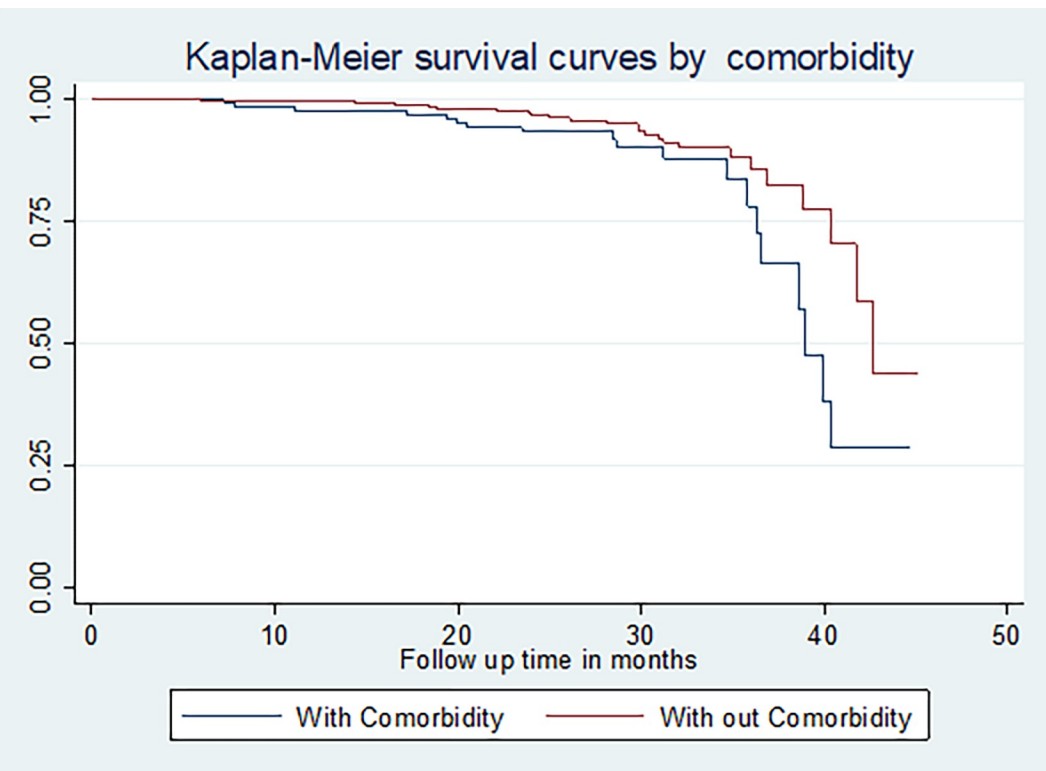

**Fig 5. The Kaplan-Meier survival curves comparing survival time of cervical cancer patients by co-morbidity (n = 422, March 2021).**

higher than in India (8.1%). [9, 17, 21, 25]. This may be due to the difference in study area and period.

The cumulative survival rates for early and late stages of disease in this study (78.9% and 56.7% respectively) are comparable with a study conducted in India (84% and 8.1% respectively) [13]. The variations may be due to the duration of follow-up time.

The one-year survival of cervical cancer patients treated by radiotherapy in this study was 90.24%. This is higher than the previous study conducted at Addis Ababa University Hospital [26], but it is in line with the study conducted in Malaysia [27]. The difference may be a result due to different study periods.

The overall survival rate of cervical cancer patients in this study varies from a previous study that was conducted at TASH, in Ethiopia [9]. This is due to different follow-up time duration.

The 3.8-year survival in this study is lower than a research finding conducted in China which was 79.8% [28]. This may be due to differences in the study area or treatment practice.

Age category, residential address, Parity, substance use, anemic status, the aim of radiological treatment, and several treatment cycles were important predictors that increased the risk of death in other studies [9, 16, 20]. This may be due to differences in study period and setting, as well as personal characteristics.

In the current study, marital status, stages of cervical cancer disease, tumor type, condition of co-morbidity, and the types of initiated treatments were significant predictors of cancer survival.

In this study, patients who had husbands were 0.15 times less likely to be at risk than those who were unmarried. This finding is in line with the study finding at a reference hospital in the Brazilian Amazon [11].

**Table 5. Restricted mean survival time, cumulative survival probability, and log rank test of cervical cancer patients at Felege Hiwot Comprehensive Specialized Hospital Oncology Center, Ethiopia (N = 422, March 2021).**

| Covariates | RMST, In months (95% CL) | Overall 3.8-year Survival (%) | p-value with the log-rank test |
|---|---|---|---|
| Age at diagnosis | | | 0.30 |
| <29 years | 38.11 (33.80, 42.42) | 71.37 | |
| 30–39 years | 38.95 (36.29, 41.60) | 77.19 | |
| 40–49 years | 40.71 (38.04, 43.39) | 54.61 | |
| 50–59 years | 41.36 (39.15, 43.58) | 61.16 | |
| > = 60 years | 38.85 (37.24, 40.46) | 36.74 | |
| No of children | | | **< 0.01*** |
| No child | 34.12 (30.58, 37.66) | 30.53 | |
| One child | 41.36 (39.36, 43.35) | 64.65 | |
| Two children | 40.36 (38.71, 42.00) | 55.74 | |
| Three or more children | 39.69 (37.47, 41.92) | 46.78 | |
| Residence | | | **0.03*** |
| Urban | 41.93 (40.13, 43.74) | 72.73 | |
| Rural | 39.37 (37.85, 40.88) | 44.89 | |
| Marital status | | | **< 0.01*** |
| Married | 41.86 (40.42, 43.32) | 64.10 | |
| Unmarried | 34.91 (31.54, 38.29) | 18.24 | |
| Divorced | 37.84 (35.41, 40.28) | 48.88 | |
| Widowed | 37.02 (34.66, 39.37) | 40.27 | |
| Substance Use | | | 0.07 |
| Yes | 38.65 (36.23, 41.06) | 41.80 | |
| No | 40.53 (39.16, 41.90) | 57.41 | |
| FIGO stage | | | **0.01*** |
| Stage I | 41.94 (40.00, 43.88) | 61.89 | |
| Stage II | 40.16 (38.59, 41.72) | 72.36 | |
| Stage III | 39.22 (37.38, 41.05) | 43.78 | |
| Stage IV | 36.57 (34.30, 38.83) | 9.18 | |
| Histology type | | | **< 0.01*** |
| Squamous cell carcinoma | 41.05 (39.75, 42.36) | 58.71 | |
| Adenocarcinoma | 34.90 (32.11, 38.00) | 12.38 | |
| Level of anemia | | | 0.28 |
| Anemia | 39.46 (37.93, 40.99) | 57.91 | |
| No anemia | 40.54 (38.85, 42.23) | 49.48 | |
| Co-morbidity | | | **0.03*** |
| Yes | 38.14 (36.05, 40.22) | 33.95 | |
| No | 41.17 (39.71, 42.64) | 62.87 | |
| Initial treatment initiated | | | **< 0.01*** |
| Chemotherapy | 40.58 (39.02, 42.14) | 64.73 | |
| Radiation | 33.48 (27.60, 39.35) | 25.17 | |
| Surgery | 35.46 (30.55, 40.37) | 16.40 | |
| Combination of treatments | 40.83 (39.28, 42.37) | 59.64 | |
| Combination of treatments | | | 0.61 |
| Surgery + chemotherapy | 37.16 (35.03, 39.30) | 60.00 | |
| Chemoradiation only | 40.44 (38.64, 42.23) | 51.87 | |
| surgery & radiation only | 38.64 (37.39, 39.90) | 68.18 | |
| Surgery + chemotherapy + radiation | 34.85 (27.65, 42.06) | 57.14 | |
| No combined treatments | 39.65 (37.69, 41.62) | 46.89 | |

*(Continued)*

**Table 5.** (Continued)

| Covariates | RMST, In months (95% CL) | Overall 3.8-year Survival (%) | p-value with the log-rank test |
|---|---|---|---|
| Aim of treatment | | | 0.41 |
| Radical | 39.42 (36.90, 41.94) | 56.51 | |
| Palliative | 41.02 (38.98, 43.06) | 54.89 | |
| No Radiotherapy | 39.70 (38.38, 41.01) | 51.01 | |
| No chemotherapy cycles | | | < **0.01**$^*$ |
| No chemotherapy | 38.36 (35.58, 41.14) | 45.75 | |
| First cycle | 35.39 (28.90, 41.89) | 22.67 | |
| Second cycle | 38.01 (36.00, 40.03) | 0.00 | |
| Third cycle | 35.56 (34.20, 36.94) | 35.78 | |
| Four or more cycles | 42.42 (41.10, 43.74) | 67.21 | |

$^*$ indicates the significantly associated categorical variables at p < 0.05 in Log-rank test for equality of survivor functions with 95% confidence level.

According to the current study, women with advanced disease stage (stage IV) were 6.10 times at high risk to die as compared to those with early stage of disease (FIGO stage I). This association holds but is lesser than a study conducted at Addis Ababa [9] and greater than studies in SSA [24, 29] and northwest Russia [20]. The reason may be due to the study period, the overall condition of patients during the presentation, and other different predictors.

Those women with adenocarcinoma were 0.89 times at high risk to die than those with Squamous cell carcinoma. This result is supported by the studies conducted in Nigeria [17] and Ethiopia [9].

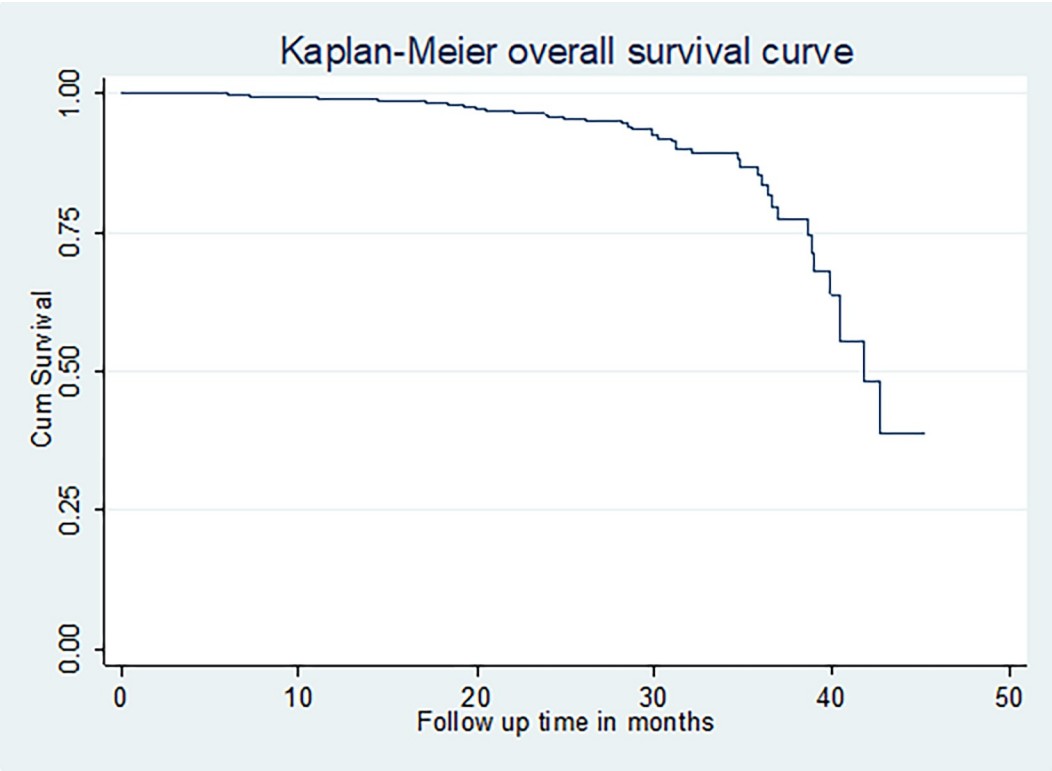

**Fig 6. Overall Kaplan-Meier estimation of the overall survival of cervical cancer patients at Felege Hiwot Comprehensive Specialized Hospital Oncology Center, Ethiopia (n = 422, March 2021).**

**Table 6. Table Schoenfeld residual test to check the assumption of proportional hazard.**

| Category | rho | chi2 | df | Prob>chi2 |
|---|---|---|---|---|
| <30 years | -0.06047 | 0.16 | 1 | 0.6899 |
| 30–39 years | 0.09744 | 0.4 | 1 | 0.5249 |
| 40–49 years | 0.04941 | 0.14 | 1 | 0.7065 |
| 50–59 years | -0.06574 | 0.23 | 1 | 0.629 |
| > = 60 years | . | . | 1 | . |
| Urban | . | . | 1 | . |
| Rural | -0.13278 | 0.77 | 1 | 0.3789 |
| Married | 0.23863 | 2.37 | 1 | 0.1235 |
| Unmarried | . | . | 1 | . |
| Divorced | -0.0461 | 0.11 | 1 | 0.7364 |
| Widowed | 0.12109 | 0.72 | 1 | 0.3964 |
| substance user | 0.09379 | 0.44 | 1 | 0.5068 |
| Not substance user | . | . | 1 | . |
| Stage I | . | . | 1 | . |
| Stage II | -0.19359 | 2.19 | 1 | 0.1392 |
| Stage III | -0.15631 | 1.64 | 1 | 0.2004 |
| Stage IV | -0.03183 | 0.05 | 1 | 0.823 |
| Squamous cell | . | . | 1 | . |
| Adenocarcinoma | 0.14597 | 0.81 | 1 | 0.3689 |
| Comorbidity | 0.00183 | 0 | 1 | 0.9897 |
| No Comorbidity | . | . | 1 | . |
| Chemotherapy | -0.23467 | 2.96 | 1 | 0.0852 |
| Radiation | -0.34803 | 7.57 | 1 | 0.0059 |
| Surgery | 0.01338 | 0.01 | 1 | 0.9128 |
| Combination of treatments | . | . | 1 | . |
| No child | -0.01509 | 0.02 | 1 | 0.9001 |
| One child | 0.01608 | 0.01 | 1 | 0.9162 |
| Two children | 0.10161 | 0.66 | 1 | 0.4172 |
| Three or more children | . | . | 1 | . |
| global test | | 17.68 | 20 | **0.6087**[*] |

[*]There was no significant evidence that suggests the assumption of proportional hazards was violated.

The current study showed that women with adenocarcinoma were 3.12 times at high risk to die than those with Squamous cell carcinoma. This is in line with a study in Russia [20].

In this study, cervical cancer patients who had a co-morbid illness were 2.57 at high risk to die than those without co-morbidity. This finding is in line with a study in Australia [30].

Cervical cancer patients who had been treated by radiotherapy for radical cure were a 1.07 times higher hazard to develop the event than those who patients initiated with a combination of treatments. This finding needs to be supported by other study findings to justify reasonable evidence of the association.

## Strength and limitations of the study

This study can reflect the current utilization of advanced treatment modalities and the overall status of cervical cancer patients.

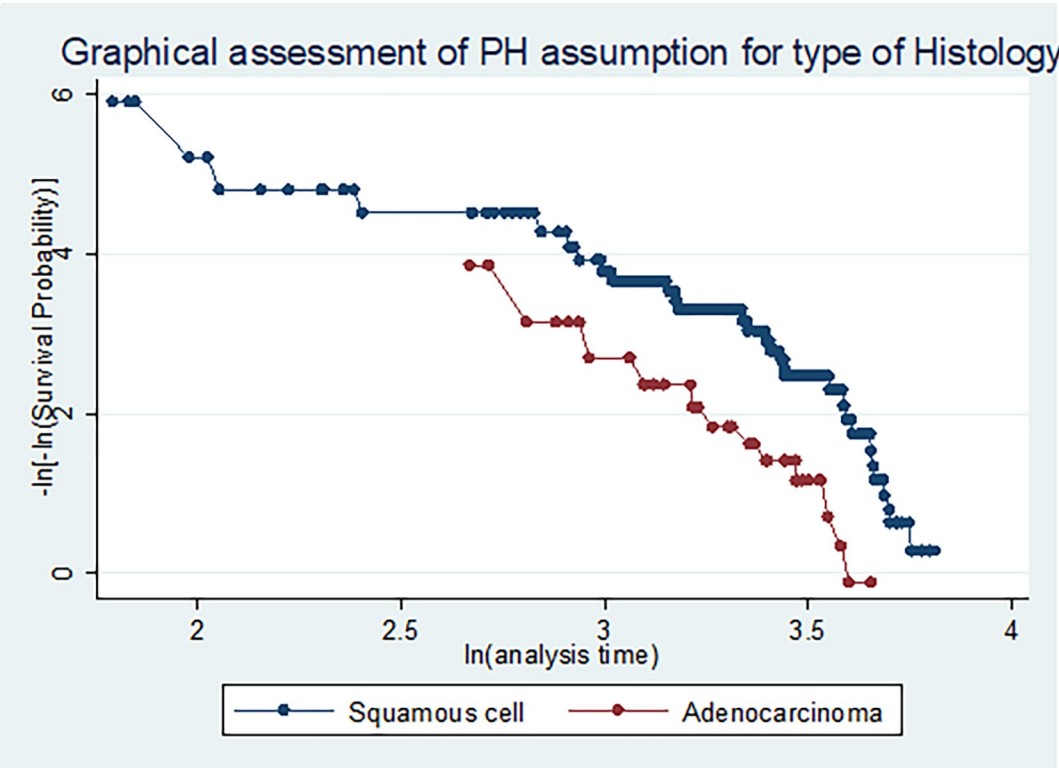

**Fig 7. Figure graphical assessment of assumption of proportional-hazards for types of histology.**

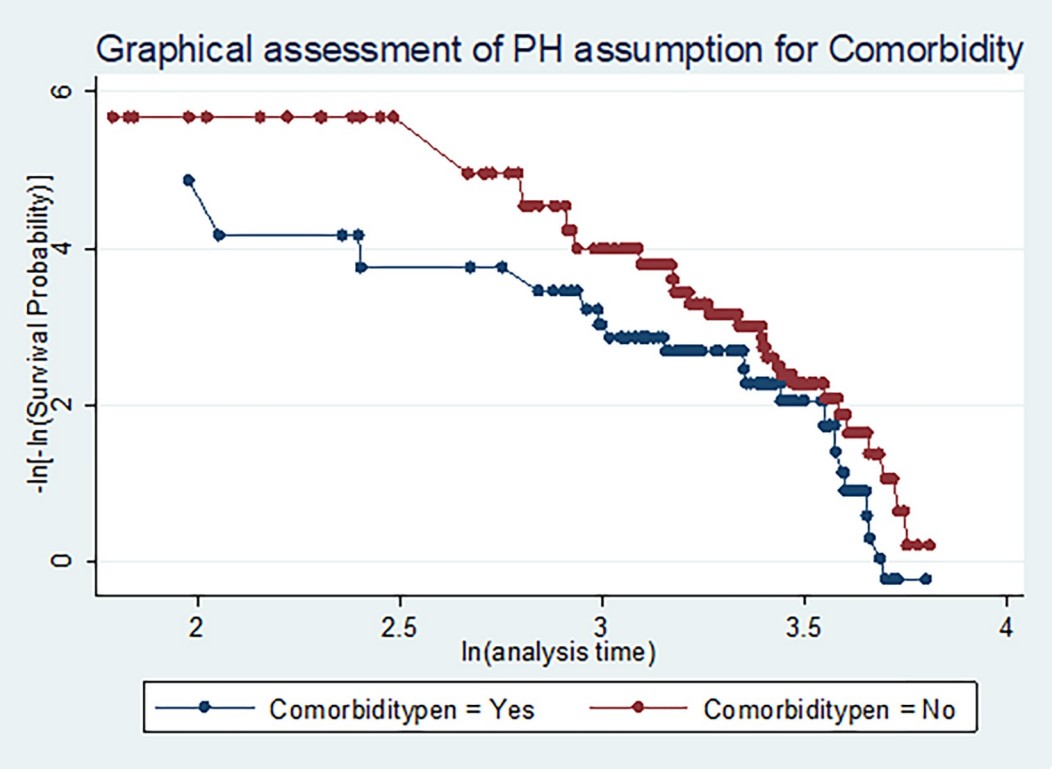

**Fig 8. Graphical assessment of assumption of proportional hazards for types of comorbidity.**

**Table 7. Results of the bivariable and multivariable Cox regression analysis of cervical cancer patients at Felege Hiwot Comprehensive Specialized Hospital Oncology Center, Ethiopia (N = 422, March 2021).**

| Covariates | Patient status, at last, Follow up | | Bivariable | Multivariable | |
|---|---|---|---|---|---|
| | Dead No (%) | Censored No (%) | CHR (95% CI) | AHR (95% CI) | p-value |
| **Age at Diagnosis** | | | | | |
| <29 years | 5 (17.86) | 23 (82.14) | 1.46 (0.53, 4.05) | 2.18 (0.66, 7.19) | 0.20 |
| 30–39 years | 6 (11.32) | 47 (88.68) | 0.93 (0.36, 2.39) | 0.73 (0.23, 2.31) | 0.59 |
| 40–49 years | 7 (5.56) | 119 (94.44) | 0.49 (0.20, 1.21) | 0.49 (0.17, 1.40) | 0.18 |
| 50–59 years | 10 (9.18) | 99 (90.82) | 0.65 (0.29, 1.45) | 0.82 (0.34, 1.99) | 0.66 |
| > = 60 years | 15 (14.15) | 91 (85.85) | 1 | 1 | |
| **Residence** | | | | | |
| Urban | 7 (5.43) | 122 (94.5) | 1 | 1 | |
| Rural | 36 (12.2) | 257 (87.7) | 2.43 (1.08, 5.47) | 2.1 (0.81, 5.22) | 0.13 |
| **No of children** | | | | | |
| No child | 12 (37.5) | 20 (62.5) | 2.96 (1.33, 6.58) | 1.52 (0.52, 4.42) | 0.44 |
| One child | 9 (7.32) | 114 (92.68) | 0.70 (0.30, 1.65) | 0.46 (0.18, 1.15) | 0.10 |
| Two children | 9 (6.47) | 130 (93.53) | 0.71 (0.30, 1.66) | 0.98 (0.36, 2.67) | 0.97 |
| > = 3 children | 13 (10.16) | 115 (89.84) | 1 | 1 | |
| **Marital status** | | | | | |
| Married | 16 (5.84) | 258 (94.1) | 0.22 (0.10, 0.48) | 0.15 (0.06, 0.37) | **0.001**\* |
| Divorced | 9 (15.52) | 49 (84.45) | 0.64 (0.26, 1.57) | 0.55 (0.18, 1.65) | 0.29 |
| Widowed | 8 (15.09) | 45 (84.91) | 0.75 (0.29, 1.90) | 0.33 (0.10, 1.02) | 0.06 |
| Unmarried | 10 (27.0) | 27 (72.97) | 1 | 1 | |
| **FIGO stage** | | | | | |
| Stage I | 8 (11.43) | 62 (88.57) | 1 | 1 | |
| Stage II | 7 (8.33) | 77 (91.67) | 1.08 (0.39, 3.00) | 2.29 (0.72, 7.23) | 0.16 |
| Stage III | 10 (8) | 115 (92) | 1.80 (0.70, 4.62) | 2.12 (0.64, 7.07) | 0.22 |
| Stage IV | 18 (12.59) | 125 (87.41) | 3.50 (1.48, 8.32) | 6.07 (2.18, 16.90) | **0.001**\* |
| **Histology type** | | | | | |
| SCC | 30 (8.09) | 341 (91.91) | 1 | 1 | |
| ADC | 13 (25.5) | 38 (74.5) | 3.97 (2.05, 7.69) | 3.12 (1.34, 7.27) | **0.008**\* |
| **Comorbidity** | | | | | |
| Yes | 19 (14.62) | 111 (85.38) | 1.95 (1.07, 3.57) | 2.57 (1.29, 5.11) | **0.007**\* |
| No | 24 (8.22) | 268 (91.78) | 1 | 1 | |
| **Substance use** | | | | | |
| Yes | 5 (15.15) | 28 (84.85) | 1.79 (0.95, 3.37) | 1.78 (0.81, 3.91) | 0.15 |
| No | 38 (9.78) | 351 (90.23) | 1 | 1 | |
| **Initial treatment** | | | | | |
| Chemotherapy | 8 (6.78) | 110 (93.22) | 0.82 (0.36, 1.84) | 0.86 (0.34, 2.16) | 0.75 |
| Radiation | 8 (34.78) | 15 (65.22) | 3.69 (1.59, 8.54) | 4.57 (1.60, 13.06) | **0.005**\* |
| Surgery | 6 (54.55) | 5 (45.45) | 3.46 (1.35, 8.89) | 0.90 (0.24, 3.41) | 0.87 |
| Combination of treatments | 21 (7.78) | 249 (92.22) | 1 | 1 | |

\* indicates the variables significantly associated with the outcome variable at < 0.05 in multivariable analysis with a 95% confidence level.

As a limitation, this study used secondary data sources from patients' abstract book by which some variables like religion, history of oral contraception use, income and occupational status were not recorded. As well as this research did not include any information about HPV vaccination in Ethiopia due to unavailability of documented source.

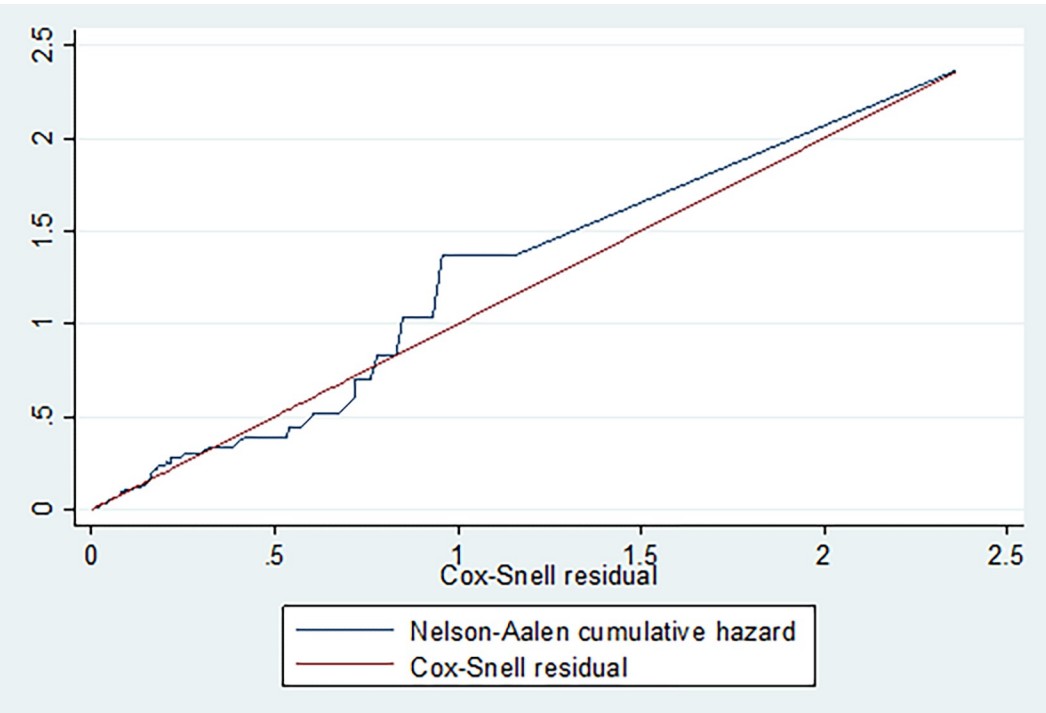

**Fig 9. Cox-Snell residual to check proportional hazard model fitness.**

## Conclusions

This study documented the time to death and the predictors of mortality among cervical cancer patients. While the median survival time was undetermined, the restricted mean survival time of cervical cancer patients in this study was 40.21 months.

The overall probability of survival rate among cervical cancer patients was 53.15%, which was lower when compared with those of high- and middle-income countries. Marital status, type of histology, stage of disease, type of treatment initiated, and presence of co-morbidity were statistically significant predictors of cervical cancer mortality. Expanding cervical cancer early screening programs, cervical cancer treatment facilities and reducing substance abuse play a key role in reducing maternal morbidity and mortality from cervical cancer in line with 90-7-90 WHO strategy for elimination of CC. In addition to this, timely reporting of advanced cervical cancer is recommended to maximize the survival time of patients with cervical cancer.

## Acknowledgments

We would like to thank Debre Markos University and FHCSH for their permission to this research and we gratefully acknowledge all data collectors and supervisors.

## Author Contributions

**Conceptualization:** Andamlak Eskale Mebratie, Nurilign Abebe Moges.

**Data curation:** Andamlak Eskale Mebratie, Belsity Temesgen Meselu.

**Formal analysis:** Andamlak Eskale Mebratie, Nurilign Abebe Moges, Belsity Temesgen Meselu.

**Investigation:** Andamlak Eskale Mebratie, Belsity Temesgen Meselu.

**Methodology:** Andamlak Eskale Mebratie, Nurilign Abebe Moges, Belsity Temesgen Meselu.

**Project administration:** Andamlak Eskale Mebratie, Nurilign Abebe Moges, Belsity Temesgen Meselu.

**Software:** Belsity Temesgen Meselu, Misganaw Fikrie Melesse.

**Supervision:** Nurilign Abebe Moges, Belsity Temesgen Meselu.

**Validation:** Nurilign Abebe Moges, Misganaw Fikrie Melesse.

**Writing – original draft:** Belsity Temesgen Meselu, Misganaw Fikrie Melesse.

**Writing – review & editing:** Misganaw Fikrie Melesse.

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
