## [Decision Letter · Decision Letter 0]

11 May 2022

PONE-D-22-09623Time to death and predictors among cervical cancer patients in Felege Hiwot Comprehensive Specialized Hospital, North West Ethiopia: Facility-based retrospective follow-up studyPLOS ONE

Dear Dr. Melesse,

Thank you for submitting your manuscript to PLOS ONE. After careful consideration, we feel that it has merit but does not fully meet PLOS ONE’s publication criteria as it currently stands. Therefore, we invite you to submit a revised version of the manuscript that addresses the points raised during the review process. Please review the external reviewer's comments and modify your manuscript as appropriate.

We look forward to receiving your revised manuscript.

Kind regards,

James P Brody

Academic Editor

PLOS ONE

Journal Requirements:

3. Please ensure that you refer to Figure 1 in your text as, if accepted, production will need this reference to link the reader to the figure.

4. Please upload a copy of Figure 10, to which you refer in your text on pages 24 and 27. If the figure is no longer to be included as part of the submission please remove all reference to it within the text.

Reviewers' comments:

Reviewer's Responses to Questions

**Comments to the Author**

1. Is the manuscript technically sound, and do the data support the conclusions?

Reviewer #1: Partly

2. Has the statistical analysis been performed appropriately and rigorously? 

Reviewer #1: I Don't Know

3. Have the authors made all data underlying the findings in their manuscript fully available?

Reviewer #1: Yes

4. Is the manuscript presented in an intelligible fashion and written in standard English?

Reviewer #1: No

5. Review Comments to the Author

Reviewer #1: 1. Title of the study not written clearly. Predictors of what?

2. First time abbreviation usage in the article not explained in a bracket but rather put at the end of the article.

3. In the abstract "being initiated with radiotherapy" is mentioned to increase the risk of death from cervical cancer, This is not clear to readers. Is the treatment really increasing the risk of death or is there another cofactor with those patients receiving radiotherapy? Otherwise this is technically/scientifically unsound.

4. The conclusion drawn is not coherent with the data presented. The data did not presented about Treatment of comorbidities yet it is stated as "Treatment of comorbidities in the early stage of cervical cancer plays a

key role in maximizing the survival time of cervical cancer patients."

5. The Language used requires editing, "TARH" in the introduction section and "radiological treatment" in the discussion part are some of the words to mention.

6. PLOS authors have the option to publish the peer review history of their article (what does this mean?). If published, this will include your full peer review and any attached files.

Reviewer #1: No

---

## [Author Response · Author response to Decision Letter 0]

23 May 2022

We accept all the comments given by both the editor and the reviewer. We would like to thank you for your thoughtful and interesting comments. We have tried to address all the comments and concerns as much as we can.

---

## [Editor Report · Decision Letter 1]

24 May 2022

Time to death from cervical cancer and predictors among cervical cancer patients in Felege Hiwot Comprehensive Specialized Hospital, North West Ethiopia: Facility-based retrospective follow-up study

PONE-D-22-09623R1

Dear Dr. Melesse,

We’re pleased to inform you that your manuscript has been judged scientifically suitable for publication and will be formally accepted for publication once it meets all outstanding technical requirements.

Kind regards,

James P Brody

Academic Editor

PLOS ONE
---

## [Editor Report · Acceptance letter]

16 Jun 2022

PONE-D-22-09623R1 

Time to death from cervical cancer and predictors among cervical cancer patients in Felege Hiwot Comprehensive Specialized Hospital, North West Ethiopia: Facility-based retrospective follow-up study 

Dear Dr. Melesse:

I'm pleased to inform you that your manuscript has been deemed suitable for publication in PLOS ONE. Congratulations! Your manuscript is now with our production department. 

Kind regards, 

on behalf of

Dr. James P Brody 

Academic Editor

PLOS ONE